# Wound Healing Efficacy of Cucurbitaceae Seed Oils in Rats: Comprehensive Phytochemical, Pharmacological, and Histological Studies Tackling AGE/RAGE and Nrf2/Ho-1 Cue

**DOI:** 10.3390/ph17060733

**Published:** 2024-06-05

**Authors:** Ayat M. Emad, Engy A. Mahrous, Dalia M. Rasheed, Fatma Alzahraa M. Gomaa, Ahmed Mohsen Elsaid Hamdan, Heba Mohammed Refat M. Selim, Einas M. Yousef, Hagar B. Abo-Zalam, Amira A. El-Gazar, Ghada M. Ragab

**Affiliations:** 1Pharmacognosy Department, Faculty of Pharmacy, October 6 University, Sixth of October City 12585, Egypt; ayatEmad@o6u.edu.eg (A.M.E.); daliarasheed@o6u.edu.eg (D.M.R.); 2Pharmacognosy Department, Faculty of Pharmacy, Cairo University, Kasr El-Aini St., Cairo 11562, Egypt; engy.abdelhamid@pharma.cu.edu.eg; 3Microbiology and Immunology, Faculty of Pharmacy, Al-Baha University, Al Baha 65511, Saudi Arabia; fgomaa@bu.edu.sa; 4Microbiology and Immunology Department, Faculty of Pharmacy (Girls), Al-Azhar University, Cairo 35527, Egypt; 5Department of Pharmacy Practice, Faculty of Pharmacy, University of Tabuk, Tabuk 71491, Saudi Arabia; a_hamdan@ut.edu.sa; 6Department of Pharmaceutical Sciences, College of Pharmacy, AlMaarefa University, Diriyah, Riyadh 11597, Saudi Arabia; hmustafa@um.edu.sa; 7Histology and Cell Biology Department, Faculty of Medicine, Menoufia University, Shebin Elkom 3251, Egypt; enasesawy@gmail.com; 8Pharmacology and Toxicology Department, Faculty of Pharmacy, October 6 University, Sixth of October City 12585, Egypt; hagar_belal@o6u.edu.eg; 9Pharmacology and Toxicology Department, Faculty of Pharmacy, Misr University for Science & Technology (MUST), Sixth of October City 12585, Egypt; ghada.ragab@must.edu.eg

**Keywords:** Cucurbitaceae seeds, linoleic acid, wound healing, EGF, CX43, Nrf2

## Abstract

The Cucurbitaceae family includes several edible species that are consumed globally as fruits and vegetables. These species produce high volumes of seeds that are often discarded as waste. In this study, we investigate the chemical composition and biological activity of three seed oils from Cucurbitaceae plants, namely, cantaloupe, honeydew, and zucchini, in comparison to the widely used pumpkin seed oil for their ability to enhance and accelerate wound healing in rats. Our results showed that honeydew seed oil (HSO) was effective in accelerating wound closure and enhancing tissue repair, as indicated by macroscopic, histological, and biochemical analyses, as compared with pumpkin seed oil (PSO). This effect was mediated by down-regulation of the advanced glycation end products (AGE) and its receptor (RAGE) cue, activating the cytoprotective enzymes nuclear factor erythroid 2 (Nrf2) and heme oxygenase-1 (HO-1), suppressing the inflammatory mediators tumor necrosis factor (TNF)-α, nuclear factor kappa B (NF-κB), and nod-like receptor protein 3 (NLRP3), and reducing the levels of the skin integral signaling protein connexin (CX)-43. Furthermore, immunohistochemical staining for epidermal growth factor (EGF) showed the lowest expression in the skin after treatment with HSO, indicating a well-organized and complete healing process. Other seed oils from cantaloupe and zucchini exhibited favorable activity when compared with untreated rats; however, their efficacy was comparatively lower than that of PSO and HSO. Gas chromatographic analysis of the derivatized oils warranted the superior activity of HSO to its high nutraceutical content of linoleic acid, which represented 65.9% of the fatty acid content. This study’s findings validate the use of honeydew seeds as a wound-healing fixed oil and encourage further investigation into the potential of Cucurbitaceae seeds as sources of medicinally valuable plant oils.

## 1. Introduction

Wound healing is a complicated and well-organized multistep process that comprises a series of overlapping phases, including hemostasis, inflammation, proliferation, and remodeling [1]. Nevertheless, this cascade of events can be interrupted, resulting in a delay in the healing process of acute wounds or the formation of persistent nonhealing chronic wounds. Both chronic and slowly healing acute wounds are distinguished by an excessive increase in leucocytes, heightened levels of cytokines and chemokines, and, as a result, increased breakdown of the structural components of tissues [2].

Inflammation is implicated as a causative factor in delayed healing, and data suggest that in the absence of infection, the inflammatory response may be severe [3]. During instances of inflammation-related pathology, the receptor for advanced glycation end products (RAGE) is a notable target because of its role in regulating gene transcription in inflammation and re-epithelialization [4]. This membrane receptor is a multiligand receptor in the immunoglobulin superfamily. It binds alarmins, and one of them is advanced glycation end products (AGEs) [5]. The activation of the RAGE receptor triggers the activation of several intracellular signaling pathways through multiple mechanisms. These mechanisms include regulating the release of cytokines and chemokines, promoting cell-to-cell adhesion, and stimulating the synthesis of matrix metalloproteases and cytoskeletal proteins [6,7].

A wide range of treatment approaches and modalities have been utilized, such as swabbing to detect infection, removing tissue debris from the wound bed, performing transplantation and cell therapy, providing wound dressing, and employing instrumental techniques [7]. Plant-derived natural compounds have the potential to enhance the wound-healing process by mitigating scar formation with antimicrobial, antioxidant, and healing properties that promote blood coagulation and accelerate wound healing. Several phytochemicals viz. curcumins, catechins, and silymarin have been widely recognized with potent wound healing efficacy both in vitro and in vivo [8]. These compounds are relatively safe in comparison with synthetic substances, as they are better tolerated and can be significantly more cost-effective than conventional therapies [9].

Cucurbitaceae (gourd family) is a large plant family that includes several species with edible fruits such as melons, squash, pumpkin, and cucumber. Cucurbitaceae edible fruits produce a large volume of seeds that are discarded as organic waste. Recently, many investigators tried to exploit Cucurbitaceae seeds as potential sources of plant oils encouraged by a high yield that can reach more than 50% [10]. It has been shown that the bioactive compounds in Cucurbitaceae seed oils are a suitable source of nutraceuticals and can be denoted as health-promoting compounds [11]. Among Cucurbitaceae seed oils, pumpkin seed oil (PSO) has been repeatedly shown to accelerate wound healing in rats [12].

Given the general trend of using natural products in treatment strategies, it is necessary to gather further data to evaluate the wound healing effectiveness of such products, especially seed oils. This approach may have the potential for utilizing seeds to yield advantages in both industry and environmental conservation. Therefore, we chose economically feasible sources of seed oils that are widely available and of low cost. Our study aims to evaluate the wound healing potentials of these seed oils, explore the possible molecular mechanism behind any observed activity, and attempt to correlate this activity to the chemical composition of the oil. In this study, we probed the potential wound healing ability of the oils obtained from three common Cucurbitaceae fruit seeds viz. cantaloupe (*Cucumis melo* L. var. cantalupensis), honeydew (*Cucumis melo* L., Inodorus Group), and zucchini (*Cucurbita pepo* var. giromontiina) on experimental rats. Their seed oils were also categorized based on their chemical compositions and wound healing activity by comparison to the medicinally reputed pumpkin seed oil (*Cucurbita pepo* L.).

## 2. Results

### 2.1. Chemical Composition of Cucurbitaceae Seed Oils

The seeds investigated in this study gave a relatively high yield of fixed oil especially from *Cucurbita sp.* (37.8 and 35% for PSO and ZSO, respectively), while yield was calculated at 24.8 and 23.2% for HSO and CSO, respectively. Unsaturated fatty acids were detected in high percentage in FAMEs ranging from 77.29% in PSO to 83.65% for HSO, as shown in Table 1. Linoleic acid was the major fatty acid in all investigated seeds with the highest percentage (65.9%) detected in HSO.

Analysis of the unsaponifiable matter of the oils (Table 2) indicated significant amounts of plant sterols and triterpenes except for ZSO. PSO was characterized by a high content of squalene at 54.12%, while both HSO and CSO showed significant concentrations of stigmasterol and β-sitosterol at approximately 50% of the unsaponifiable matter, as shown in Table 2.

### 2.2. Macroscopic Quantitative Assessment of Wound Healing Progression

A significant increase in the progression of wound healing (%) was observed in all treatment groups compared with the non-treated group, as shown in Figure 1A–C, albeit to variable degrees depending on the type of treatment. Treatment with HSO showed the fastest healing rate (group IV) when compared with all other treatment groups, especially on day 8 (88.33% ± 0.84) and day 10 (98.63% ± 0.70), even when compared with the PSO-treated group (day 8; 78.07% ± 0.75 and day 10; 94.85% ± 0.70). On the other hand, ZSO showed the slowest wound healing (day 8, 46.27% ± 0.90 and day 10, 91.0% ± 1.42) and no significant difference with the untreated wounded group at day 14.

### 2.3. Effect of Different Oils on Histopathological Alterations after Wound Induction

Skin from the normal control group demonstrated normal histology consisting of an epidermis and dermis rich in collagen with the presence of numerous hair follicles, sweat glands, and sebaceous glands, as shown in Figure 2 and Table 3. In contrast, despite the wound gap being filled with organized tissue, the wound region in the wound injury of the untreated group displayed evidence of delayed healing. On the other hand, groups treated with PSO/ST, HSO, and CSO (III, IV, and V, respectively) exhibited enhanced healing criteria when compared with the wound injury group that were not evident in animals treated with ZSO (VI).

### 2.4. Effect of Different Oils on the Protein Expression of AGE and RAGE as Signaling Cue/Genes

A marked increase in the protein expression of AGE and RAGE in the skin tissues of injured rats was observed as compared with the intact skin of the normal control group. Meanwhile, wounded rats treated with PSO/ST, HSO, CSO, or ZSO turned off the [AGE/RAGE] cue, as evidenced by decreased protein expression folds by 1.8/2.3, 2.2/2.3, 1.3/2.1, and 2/2.3, respectively (Figure 3).

### 2.5. Effect of Different Oils on the Tissue Contents of Nrf2/HO-1 as Antioxidant Signaling Molecules

Wounded rats exhibited a significant reduction in Nrf2/HO-1 skin contents as compared with the skin content of healthy uninjured rats. On the other hand, treatment with different seed oils increased the Nrf2 and HO-1 content with HSO treatment exhibiting the highest antioxidant potential among the treatment groups and even superseding values obtained after PSO/ST treatment (Figure 4).

### 2.6. Effect of Different Oils on TNF-α, NF-κB, and NLRP3 as Inflammatory Markers in Addition to CX-43 as Skin Integral Signaling Protein

A strong inflammatory response was observed in untreated animals, depicted by the remarkable increase in TNF-α (4 folds), NF-κB (2.55 folds), and NLRP3 (3.6 folds), as compared with the normal control group (Figure 5). Treatment with PSO/ST, HSO, CSO, or ZSO displayed anti-inflammatory efficacy via decreasing TNF-α, NF-κB, and NLRP3. Moreover, the skin contents of CX (connexin)-43 were significantly increased in the wound injury group compared with the normal control group by 500%. Injured rats treated with PSO/ST, HSO, CSO, and ZSO presented a significant decrease in CX-43 by 70.5%, 78.6%, 65%, and 60%, respectively, when compared with the wound-injured group. Notably, treatment with HSO showed the utmost decrease in TNF-α, NF-κB, NLRP3, and CX-43 levels (Figure 5).

### 2.7. Effect of Different Oils on Immunohistochemistry Expression of Epidermal Growth Factor (EGF) after Wound Induction

As presented in Figure 6, the wounded rats revealed a significant up-regulation in the immunohistochemistry (IHC) expression of EGF (20.80 ± 3.3) as compared with the healthy ones. However, post-application of PSO/ST and HSO significantly reduced EGF expression as compared with the wounded untreated rats by 35.14% and 60.4%, respectively. The lowest EGF expression was observed after HSO treatment, indicating a well-organized and complete healing process.

## 3. Discussion

In this study, we aimed to identify the wound healing potential of Cucurbitaceae seed oils that are efficient, widely available, and of low cost that can be obtained from organic food waste. In this respect, honeydew melon seed oil (HSO), cantaloupe seed oil (QSO), and zucchini seed oil (ZSO) were investigated for their ability to accelerate wound healing when topically applied over the wounded area after induction of excisional wound injury in Wistar rats’ back by comprehensive phytochemical, pharmacological, and histological investigations. Their effect was compared to pumpkin seed oil (PSO/ST) as a standard topical remedy, which has previously shown good wound healing capacity [12,13].

The four investigated oils showed a high content of unsaturated fatty acids in agreement with previous reports [14,15]. The most abundant fatty acid in the four investigated seed oils was linoleic acid. This finding is very significant since previous studies have shown that topical application of pure linoleic acid (30 µM) or oils containing high concentrations of linoleic acid (39–50%) can accelerate wound healing [16,17]. Furthermore, animal studies and clinical investigations indicated that linoleic acid can positively affect the wound healing process in burns, pressure ulcers, and when orally administered to diabetic animals [12,18]. Additionally, the second most abundant fatty acid in all investigated seed oils was oleic acid, which is reputed to promote wound closure by modulating the inflammatory response in the initial stages of injury [19]. Hence, the high percentage of unsaturated fatty acids in all investigated Cucurbitaceae seed oils is likely a contributing factor in their wound healing properties, especially those enriched with polyunsaturated fatty acids (linoleic and linolenic acids).

Consistent with these previous reports, the oil with the highest percentage of these fatty acids, HSO, showed the fastest wound healing rate at the initial stage of injury, as evidenced by significant wound healing observed on days 4 and 8. Meanwhile, the application of ZSO, with the lowest percentage of linoleic acid, caused the least wound healing among all treatment groups. Other constituents of these oils may contribute to the overall wound healing activity such as *β*-sitosterol and geranyl geraniol alcohol, which are reported at relatively high percentages in HSO. These constituents have been validated as possible anti-inflammatory compounds [20,21] and may be responsible for the superior wound healing activity observed for HSO. Moreover, our histological findings aligned with our macroscopic findings, where the untreated wound injury group demonstrated delayed wound healing. To understand the molecular mechanism of this wound healing ability we monitored changes in AGE/RAGE, Nrf-2/Ho-1, and NF-κB/Cx43/EGF cue.

Advanced glycation end products (AGE) are very heterogeneous substances, and their binding to RAGE may be involved in tissue regeneration and repair [7]. In the current study, the wound-injured rats exhibited a marked increase in the expression of skin AGE and RAGE as compared with the normal control animals. In contrast, the wounded rats treated with PSO/ST, HSO, CSO, and ZSO turned off AGE/RAGE protein expression. Supporting our findings, blocking the AGE-RAGE signaling cue was previously shown to be associated with improved angiogenesis, tissue granulation, and wound re-epithelialization [22].

Wound injury sites have a marked increase in reactive oxygen species (ROS) to resist pathogen invasion and attract immune cells [23]. However, prolonged ROS production may cause oxidative stress, which can hinder the healing process [24]. Therefore, the activation of nuclear factor-erythroid 2-related factor 2 (Nrf2) is protective as it regulates the transcription of cytoprotective genes, including those encoding different ROS-detoxifying enzymes, antioxidant proteins, and drug transporters [1,23]. For instance, wounded tissues activate heme oxygenase-1 (HO-1) as a protective agent against oxidative and inflammatory insults [25,26]. In addition, HO-1 is considered an important modulator of new blood vessel formation (neovascularization) [25].

In the present investigation, the animals with untreated wounds showed a significant reduction in levels of Nrf2 and HO-1, whereas post-treatment with PSO/ST, HSO, CSO, and ZSO increased the contents of Nrf2 and HO-1, providing evidence for their healing potential and ability to reduce tissue damage. Once more, the HSO treatment exhibited the most potent antioxidant potential among the other treatment groups.

The inflammatory response is momentous during the wound healing process and is a key player in tissue repair [27]. Inflammatory cells and macrophages initiate the development of granulation tissue and release a variety of pro-inflammatory cytokines, interleukins, and growth factors such as epidermal growth factor (EGF), that attract fibroblasts, stimulate keratinocyte migration and proliferation, and initiate the formation of new blood vessels [28]. However, an exaggerated, unbalanced inflammatory sequel has a direct impact on a delayed or poorly healed one. Furthermore, an increase in the inflammatory mediator NF-κB further stimulates the up-regulation of the nod-like receptor protein 3 (NLRP3) inflammasome [29].

In the current study, the inflammatory reaction was clearly progressing in the untreated wounded group. This progress was depicted by a remarkable increase in TNF-α and NF-κB. The latter stimulated an up-regulation of NLRP3, a family of cytosolic multi-proteins that helps boost cytokine maturation and release pro-inflammatory cytokines [29]. However, post-treatment with PSO/ST, HSO, CSO, and ZSO displayed anti-inflammatory efficacy via decreasing TNF-α, NF-κB, and NLRP3. The presence of anti-inflammatory molecules such as ω-3 fatty acids (linolenic acid) and terpenoidal compounds (e.g., squalene and geranyl geraniol) may have contributed to the weaker inflammatory response observed in the animals treated with Cucurbitaceae seed oils. For example, geranyl geraniol was found to suppress the production of TNf-α, Il-6, and Cox-2 among other pro-inflammatory mediators [21]. Meanwhile, the triterpenoidal compound squalene was shown to reduce the pro-inflammatory cytokines TNF-α, IL-1β, IL-6, and IFN-γ while enhancing the production of anti-inflammatory factors HO-1 and Nrf2, which were also reported at higher concentrations in this study [30].

Our study further investigated connexin 43 (Cx43) regulation after excisional wound creation. Cxs are a family of 21 molecules that generate hemichannels and gap junctions on the cell membrane. Hemichannel opening may be stimulated by mechanical signals associated with wound healing, inflammation, hypoxia, and oxidative stress. Early down-regulation of Cx43 is related to effective wound closure, but its up-regulation is connected to chronic wounds that do not heal [31]. In this study, wound induction was associated with a massive increase in Cx43 skin contents, which may explain the high inflammatory state observed in this group compared with the healthy, unwounded group. Augmenting our results, the opening of Cx hemichannels promotes the liberation of damage-associated molecular patterns, a group of naturally occurring molecules that play a crucial role in the development of inflammatory disorders, while its blocking results in decreased inflammation, less tissue damage, and enhanced organ function in both human and animal models [32,33,34,35]. On the other hand, the post-application of different oils reduced the skin contents of Cx43 as compared with the skin contents of untreated wounded rats, adding to their anti-inflammatory healing potentials. Moreover, inhibiting Cx43 expression by the four seed oils was mirrored on re-epithelialization histologic scores recorded herein.

Finally, epidermal growth factor (EGF) promotes wound contraction by increasing local fibroblast proliferation and migration and inducing dermal maturation [36]. Once wound closure (100% epithelialization) is accomplished, keratinocytes undergo differentiation and stratification to restore the skin’s protective barrier [28]. In this study, HSO treatment exhibited the lowest epidermal growth factor (EGF) expression in our immunohistochemical assessment, indicating a well-organized and complete healing process.

All previous measurements conclude that multiple molecular mechanisms mediate the significant wound healing activity of seed oils from Cucurbitaceae plants. This effect may be partially explained by the higher content of linoleic acid (LA), which was estimated in HSO at 65.9% compared with 39.26% in PSO. LA has been shown to suppress the release of inflammatory mediators such as IL-1b, IL-6, and VEGF from macrophages [37]. Additionally, all investigated oils contained a number of anti-inflammatory bioagents such as squalene (triterpene), geranylgeranol (diterpene), stigmasterol, and β-sitosterol (plant sterols), which have evident anti-inflammatory and tissue repair properties [20,21].

## 4. Material and Methods

### 4.1. Plant Material

Seeds from cantaloupe (*Cucumis melo* var. cantalupensis), honeydew (*Cucumis melo* L.—Inodorus Group), pumpkin (*Cucurbita pepo* L.), and zucchini (*Cucurbita pepo* var. giromontiina) were purchased from the agriculture research center (Giza, Egypt). Voucher samples from each type of seed were deposited at the herbarium of the Department of Pharmacognosy, Cairo University, numbers 26.10.2021I, 26.10.2021II, 26.10.2021III, and 26.10.2021IV, respectively.

### 4.2. Oil Extraction

Dry seeds (50 g each) were powdered and then extracted with *n*-hexane (3 × 150 mL) in dark amber containers. The solvent was evaporated under reduced pressure using rotatory evaporator (temperature not exceeding 35 °C) to obtain cantaloupe seed oil (CSO; yellowish oil, 11.6 g), honeydew seed oil (HSO; yellowish oil, 12.4 g), pumpkin seed oil (PSO; greenish oil, 18.9 g), and zucchini seed oil (ZSO; greenish oil, 17.5 g).

### 4.3. Chemical Analysis of the Oils

#### 4.3.1. Sample Derivatizations

Seed-fixed oils (2 g from each species) were refluxed with 50 mL of 5% methanolic KOH for 5 h [38]. After cooling, 100 mL water was added to the reaction and then extracted with diethyl ether to recover the unsaponifiable matter (USM). The aqueous layer was separated and acidified with HCl to release free fatty acids, and diethyl ether was repeatedly added to extract fatty acid mixtures, which were then refluxed with methanol (25 mL) and sulfuric acid (2 mL) for 4 h to give fatty acid methyl esters (FAMEs).

#### 4.3.2. Gas Chromatography Analysis

Gas chromatography analysis of the USM and FAME system was performed at Central Laboratories Network (National Research Centre, Cairo, Egypt) using an Agilent Technologies gas chromatograph (7890B) and mass spectrometer detector (5977A). The GC system was equipped with an HP-5MS column (30 m × 0.25 mm × 0.25 μm film thickness). Analyses were carried out using hydrogen as the carrier gas at a flow rate of 1.0 mL/min and a split ratio of 10:1, an injection volume of 1 µL, and the following temperature program: 240 °C; rising at 10 °C/min to 265 °C and held for 1 min; rising at 15 °C/min to 300 °C and held for 25 min. The injector and detector were held at 280 and 290 °C, respectively. Mass spectra were obtained by electron ionization (EI) at 70 eV using a spectral range of *m*/*z* 50–550 and solvent delay of 3 min. The mass temperature was 230 °C and Quad 150 °C. The identification of different constituents was determined by comparing the spectrum fragmentation pattern with those stored in Wiley and NIST Mass Spectral Library data, as described previously [39,40]. The contents of derivatized components were determined based on a percentage of peak areas relative to the summed peak area percent of all identified metabolites.

### 4.4. In Vivo Wound Healing Model

#### 4.4.1. Animals

Male Wistar rats (n = 54), weighing between 220 and 240 g and separated into 6 groups (n = 9/group), were purchased from the National Research Centre (NRC, Giza, Egypt). The rats were kept separately (one rat per cage) under standard environmental conditions such as the automated regulation of temperature, humidity, ventilation, and a 12 h light/dark cycle with unrestricted access to water and food. The experimental protocol was conducted in accordance with the Guide for the Care and Use of Laboratory Animals (NIH publication, 1996) and was approved by the Committee of Animal Ethics of October 6 University with approval No. PRE-Ph-2204019.

#### 4.4.2. Creation of Excision Wounds

Initially, the rats were anesthetized with halothane (1.5%) and an intraperitoneal injection of pentobarbital (0.5 mL/kg) [41,42]. Once the rats were sedated, their back hair was shaved, and their skin was purified using an alcohol solution and cleaned with sterile water. A circular wound with an approximate diameter of 10 mm was made using a skin biopsy tool and surgical scissors [43].

#### 4.4.3. Experiment Protocol

Six animals were left without wound injury induction and considered a normal control group (Group I). Other animals in the study were wound-injured as described and distributed over five groups (n = 6/group). Animals in the wound injury group (Group II) were left without any intervention. For animals in Group III (wound injury + PSO/ST), pumpkin seed oil (PSO) was topically applied over the wounded area (0.52 μL/mm^2^ two times/day) [16]. In Groups IV, V, and VI, wounds were topically treated with HSO, CSO, and ZSO, respectively, with the same dosage used for the PSO standard treatment group. Tri-M Strip^®^, an adhesive wound dressing, was applied softly to prevent oil leakage and to prevent animals from scratching their dorsal skin. The strip was chosen to be larger than the wound’s size, ensuring that only the pad touched the wound. Wound induction was made, and the treatment procedure started 24 h after wound induction (day 1) and lasted for 14 days. Wound healing was observed visually by taking photographs using a digital camera on days 4, 8, 10, and 14 of wound induction (Figure 7), as guided by our previous study [41]. The wound healing percent was determined using an image analyzer (Image J 2.0 software, NIH, New York, NY, USA) and considered using the next equation.
The progression of wound healing (%) = (initial wound area − current wound area)/initial wound area × 100

#### 4.4.4. Tissue Collection

On the last day of the experiment, the dorsal wounded skin area of each rat was collected after the animals’ euthanasia by cervical dislocation under deep anesthesia. The collected wounded skin area tissue from six rats was further cut into two halves; one half was homogenized in phosphate buffer for ELISA measurements, while the other half was further halved; one was immersed in RNA later solution for qRT-PCR assessment, and the other was homogenized in RIPA buffer for Western blot analysis. The remaining skin samples (n = 3/groups) were preserved in 10% neutral buffered formalin for histopathological and immunohistochemistry investigations.

#### 4.4.5. Quantitative Real-Time Polymerase Chain Reaction (RT-PCR)

The qPCR analysis was conducted using a real-time PCR system, and the expression levels of receptors of advanced glycation end products (RAGE) in the wound tissues were quantitatively determined using SYBR Premix ExTaq (Takara Bio, Inc., Dalian, China). RNA was extracted using the SV total RNA isolation system (Promega, Madison, WI, USA), and reverse transcription was performed using the superscript II reverse transcriptase kit (Invitrogen, Carlsbad, CA, USA). The primer sequences used for RAGE and β-actin (control for normalization) are listed in Table 4.

#### 4.4.6. Western Blotting (WB)

Western blotting was used to determine the protein level of advanced glycation end product (AGE); the primary antibody was obtained from Thermo Scientific Co., (Waltham, MA, USA) according to the method described previously [44,45]. Briefly, the RIPA fractions of skin homogenates were utilized for Western blot studies. Specifically, the total protein content of the skin homogenates was determined using the Bradford Protein Assay Kit from Bio Basic Inc., located in Ontario, Canada. The protein samples, with a quantity of 30–50 μg per lane, were first separated using SDS-PAGE and then deposited onto a nitrocellulose membrane. The membranes were obstructed with a 5% (*w*/*v*) solution of non-fat dry milk in Tris-buffered saline–Tween 20 (0.025 M Tris; 0.15 M NaCl; 0.05% Tween 20; pH 7.4) and then exposed to primary antibodies overnight at a temperature of 4 °C. The primary antibody was acquired from Thermo Scientific Co. (Waltham, MA, USA). The next day, the membranes were cleansed and exposed to secondary antibodies (Thermo Scientific Co., Waltham, MA, USA) for 1 h at 25 °C. Specific bands were visualized with a ChemiDocTMimaging system (Image LabTM software version 5.1, Bio-Rad Laboratories Inc., Hercules, CA, USA). The optical density (OD) of the values was standardized against β-actin.

#### 4.4.7. Determination of Tissue Antioxidant Stress Markers, Inflammatory Factors, and Connexin 43

The tissue contents of nuclear factor erythropoietin-2-related factor 2 (Nrf-2, Cat# MBS012148), heme oxygenase-1 (HO-1, Cat# MBS800757), tumor necrosis factor-alpha (TNF-α, Cat# MBS624076), nuclear transcription factor-kappa B (NF-κB, Cat# MBS267737), nod like receptor protein 3 (NLRP3, Cat# MBS645773), skin integral signaling protein, and connexin 43 (CX-43, Cat# MBS008326) were determined using MyBioSource ELISA kits (San Diego, CA, USA) according to manufacturer’s protocols. The homogenized skin tissue in phosphate buffer was used for ELISA measurements. Basically, it is a test method that uses plates to specifically identify and quantify various substances. In this technique, the concentration of the identified biomarker is directly proportional to the optical density (O.D) using microplate reader (Biotech, Burlington, VT, USA).

#### 4.4.8. Histopathology Examination

The skin tissues were fixed in 10% neutral buffered formalin then processed, embedded in paraffin, sectioned to a thickness of 5 mm, and stained with hematoxylin and eosin (H&E) to determine the histopathological alterations. The epidermal layer formation was recorded and scored in the H&E photomicrographs by means of re-epithelization and epidermal thickness [46].

#### 4.4.9. Immunohistochemistry (IHC) Examination

Paraffin sections were mounted on positively charged slides by using the avidin/biotin complex (ABC) method. Epidermal growth factor (EGF) monoclonal antibody (Cat# LS-C121770; Dil.: 1:25 was obtained from LifeSpan Biosciences (Seattle, WA, USA). Sections were incubated with the antibodies, and then the reagents were added through the vectastain ABC-horseradish peroxidase (HRP) kit (Vector laboratories, Newark, CA, USA). Marker expression was detected with peroxidase and colored with diaminobenzidine (DAB) (Sigma-Aldrich, St. Louis, MO, USA). Negative controls were included using non-immune serum in place of the primary or secondary antibodies. IHC stained sections were examined using an Olympus microscope (BX-53, Evident Corporation, Tokyo, Japan). The immunohistochemistry results were scored by calculating the percentage of the reaction area in 10 microscopic fields using Image J 1.53t, a software developed by Wayne Rasband and contributors at the National Institutes of Health in the USA.

### 4.5. Statistical Analysis

The mean and standard deviation (SD) for all the data were displayed. The normality of the data was checked using the Shapiro–Wilk test, and data of Western blot and PCR were further analyzed for homogeneity and shown a non-significant Levene’s median test (Brown-Forsythe) [47]; the *p*-values are displayed in Appendix A. Then, the data were analyzed using one-way analysis of variance (ANOVA), followed by Tukey’s post hoc test conducted with GraphPad Prism software (version 9; GraphPad Software, Inc., San Diego, CA, USA). A Mixed model ANOVA and Repeated Measure ANOVA were used to compare the progression of wound healing within groups at day 4, day 8, day 10, and day 14 from induction using IBM© SPSS© Statistics version 22 (IBM© Corp., Armonk, NY, USA). A *p*-value of less than 0.05 was used to indicate statistical significance.

## 5. Conclusions

This study records the auspicious wound healing capacity of vegetable oils from the Cucurbitaceae family. Cucurbitaceae fruits are characterized by the production of a large volume of seeds that are enriched in fixed oil. We investigated four common Cucurbitaceae seeds and found that all of them can accelerate and enhance wound healing. Among the investigated oils, honeydew melon seed oil (HSO) exerted a comparable wound healing power to the reputed and much more commonly employed oil in folk medicine, pumpkin seed oil. These oils can be incorporated either as hydrophobic vehicles or low-cost therapeutic agents that can be obtained in relatively high yield from food waste. HSO was markedly efficient in promoting the healing process, especially in the initial stages of injury, and resulted in well-organized tissue repair. The results presented here suggest that HSO’s healing properties come from its high content of fatty acids, especially linoleic acid, which has important anti-inflammatory properties. Further investigations are needed to better understand the mechanism of action of HSO and its bioactive compounds in enhancing tissue repair. It is worth mentioning that other Cucurbitaceae seed oils have substantial wound healing activity, and further studies are required to investigate their potential as wound healing natural remedies. Future perspectives also include formulating these oils in convenient pharmaceutical preparations, with its promotion as a nutraceutical candidate for resistant chronic wounds and clinical trials.

Limitation of this study: Despite the comprehensive results reached by this study, some limitations were identified that can be overcome in future research with similar objectives, such as using advanced techniques including molecular networking and sample size analysis techniques to select the correct sample size, and formulating the investigated oils into suitable pharmaceutical dosage form for convenient application.

## Figures and Tables

**Figure 1 pharmaceuticals-17-00733-f001:**
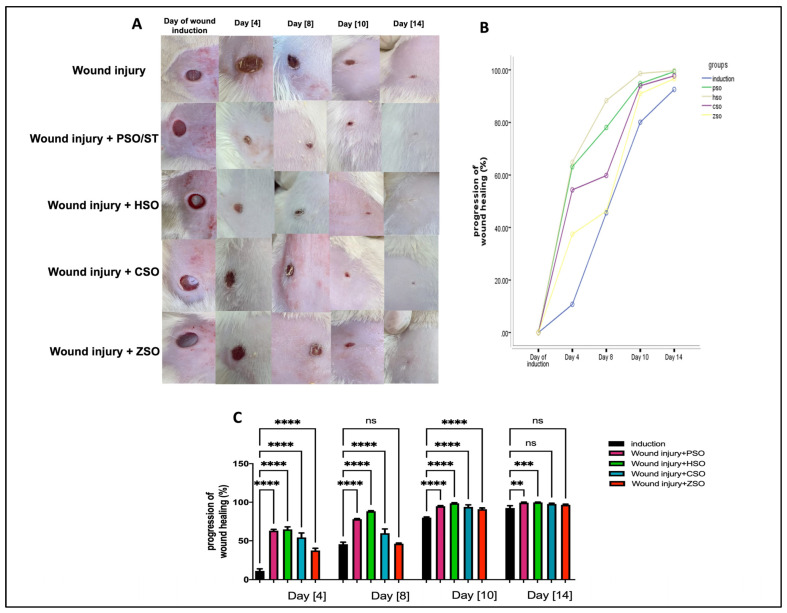
Macroscopic assessment of the morphological appearance and progression of wound healing (%) on days 4, 8, 10, and 14 after wound induction. (**A**): Representative photo-macrographs of wound morphology, (**B**): progression of wound healing (%) determined using Image J, an image analyzer, and compared using Mixed model ANOVA, (**C**): progression of wound healing (%) determined using Image J, an image analyzer, and compared using one-way ANOVA and subsequent multiple comparisons using Tukey’s test. Data are expressed as the mean ± SD [n = 9], and the number of asterisks “*” above the columns indicates the strength of significance as follows: ** *p* < 0.01, *** *p* < 0.001, and **** *p* < 0.0001, while (ns) stands for non-significant. CSO; cantaloupe seed oil, HSO; honeydew melon seed oil, PSO; pumpkin seed oil, ST; standard, ZSO; zucchini seed oil.

**Figure 2 pharmaceuticals-17-00733-f002:**
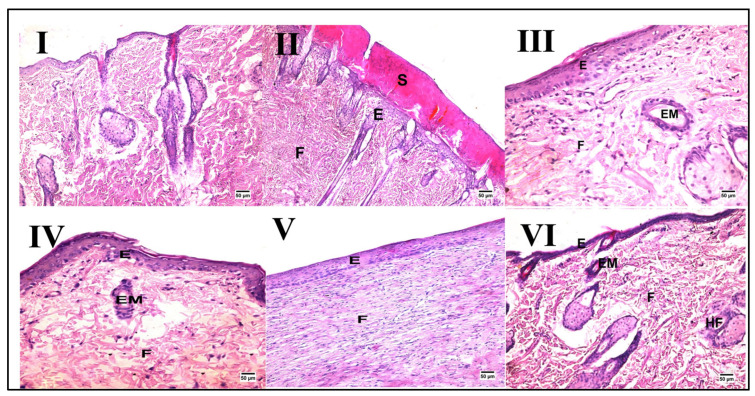
Representative photomicrographs of histopathological alterations after wound induction (hematoxylin and eosin stain; scale bar 50 μm, 200×). Photomicrographs of skin tissue from the normal control group (**I**) with the normal histological structure of the skin, the wound injury group (**II**) showing the formation of a large scab (S) covering some newly formed epidermal layers (E) with fibrous connective tissue formation (F), the PSO/ST-treated group (**III**) showing migration of epidermal cells (EM) and formation of some layers of epidermis (E) and well-organized fibrous connective tissue (F), the HSO-treated group (**IV**) presenting migration of epidermal cells (EM) and formation of some layers of epidermis (E), the CSO-treated group (**V**) showing well-organized fibrous connective tissue (F) and an epidermal layer (E) at the site of the defect, and finally, the ZSO-treated group (**VI**) showing migration of epidermal cells (EM) and formation of some layers of epidermis (E) with the presence of ill-organized fibrous connective tissue (F).

**Figure 3 pharmaceuticals-17-00733-f003:**
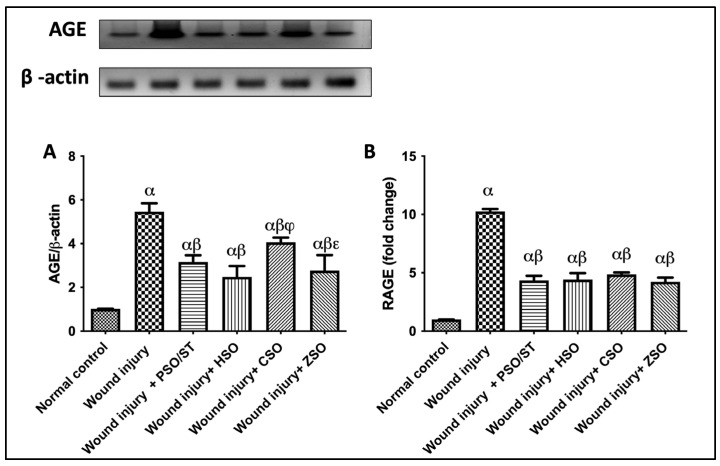
Effect of different oils on the skin tissue protein expressions of (**A**) AGE and (**B**) RAGE as signaling cue/genes in wound-injured rats. Statistical analysis was achieved with one-way ANOVA and subsequent multiple comparisons using Tukey’s test. Data are expressed as the mean ± SD [n = 3] with *p*-value < 0.05 as compared with the normal control group (α), wound injury (β), wound injury + HSO (φ), and wound injury + CSO (ε). [F-value of AGE = 60.62 and F-value of RAGE = 224.1.] AGE; advanced glycation end products, CSO; cantaloupe seed oil, HSO; honeydew melon seed oil, PSO; pumpkin seed oil, RAGE; receptor of advanced glycation end products, ST; standard, ZSO; zucchini seed oil.

**Figure 4 pharmaceuticals-17-00733-f004:**
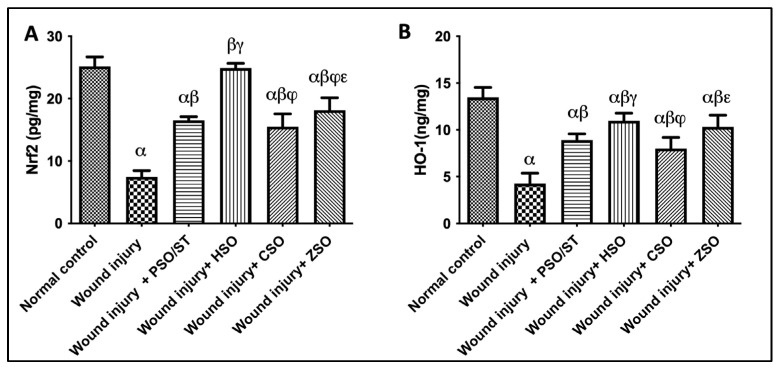
Effect of different oils on the skin tissue contents of (**A**) Nrf2 and (**B**) HO-1 as antioxidant signaling molecules in wound-injured rats. Statistical analysis was performed using one-way ANOVA and subsequent multiple comparisons using Tukey’s test. Data are expressed as the mean ± SD [n = 6] with *p*-value < 0.05 as compared with the normal control group (α) and the wound injury (β), wound injury + PSO/ST (γ), wound injury + HSO (φ), and wound injury + CSO (ε) groups. [F-value of Nrf2 = 127.9 and F-value of HO-1 = 53.09.] CSO; cantaloupe seed oil, HO-1; heme oxygenase-1, HSO; honeydew melon seed oil, Nrf2, nuclear factor erythropoietin-2-related factor 2, PSO; pumpkin seed oil, ST; standard, ZSO; zucchini seed oil.

**Figure 5 pharmaceuticals-17-00733-f005:**
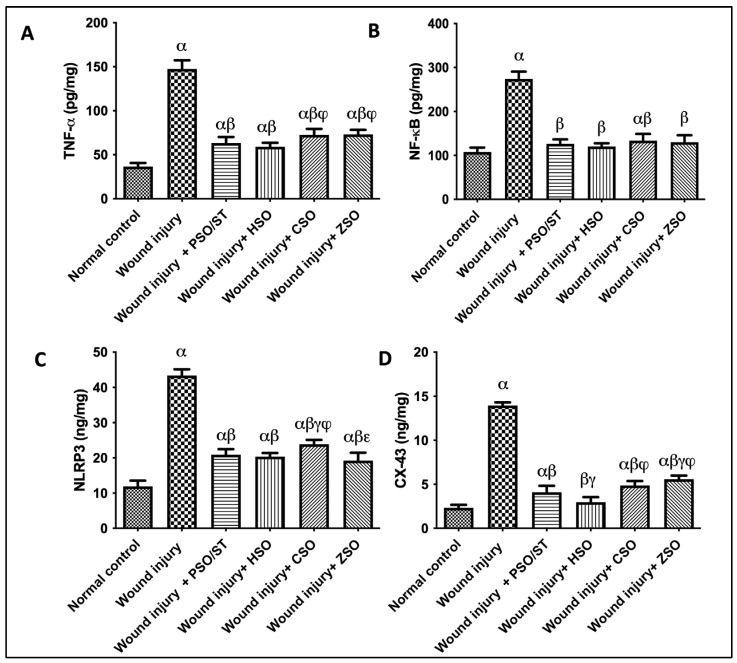
Effect of different oils on the skin tissue contents of (**A**) TNF-α, (**B**) NF-κB, and (**C**) NLRP3 as inflammatory markers in addition to serum levels of (**D**) CX-43 as a skin integrity marker in wound-injured rats. Statistical analysis was achieved with one-way ANOVA and subsequent multiple comparisons using Tukey’s test. Data are expressed as the mean ± SD [n = 6] with *p*-value < 0.05 as compared with the normal control group (α) and the wound injury (β), wound injury + PSO/ST (γ), wound injury + HSO (φ), and wound injury + CSO (ε) groups. [F-value of TNF-α = 198.9, F-value of NF-κB = 139.8, F-value of NLRP3 = 253.5, and F-value of CX-43 = 423.8.] CX-43; connexin-43, CSO; Cantaloupe seed oil, HSO; Honeydew melon seed oil, NLRP3; NLR family pyrin domain containing 3, NF-κB; nuclear transcription factor kappa B, PSO; pumpkin seed oil, ST; standard, TNF-α; tumor necrosis factor-alpha, ZSO; Zucchini seed oil.

**Figure 6 pharmaceuticals-17-00733-f006:**
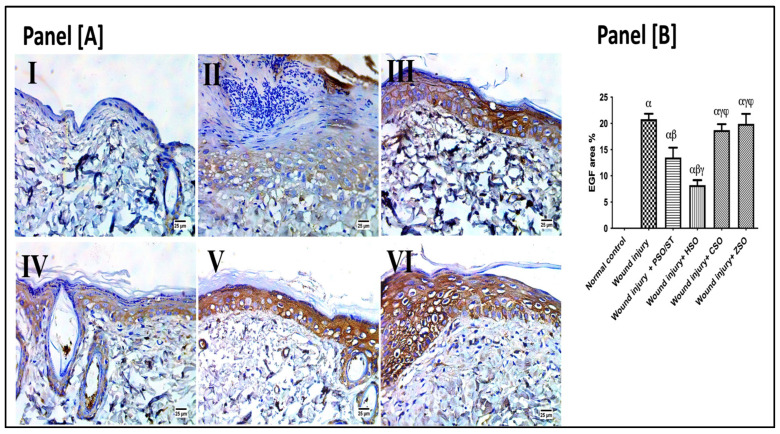
Representative photomicrographs of the immunohistochemistry expression of epidermal growth factor (EGF) after 14 days of wound induction. Panel [**A**]: skin sections of the newly formed epidermal layer showing areas stained for EGF expression in the normal skin of healthy rats (I), wound injury group (II), PSO/ST-treated rats (III), rats treated with HSO (IV), CSO (V), and ZSO (VI). Panel [**B**] represents EGF area% expression (the mean of 10 microscopic fields ± SD) in all treated groups. One-way ANOVA and subsequent multiple comparisons using Tukey’s test as compared with the normal control group (α) and the wound injury (β), wound injury + PSO/ST (γ), and wound injury + HSO (φ), groups [F-value of EGF% = 197.3]. CSO; cantaloupe seed oil; EGF; epidermal growth factor; IHC; immunohistochemistry; HSO; honeydew melon seed oil; PSO; pumpkin seed oil; ST; standard; ZSO; zucchini seed oil.

**Figure 7 pharmaceuticals-17-00733-f007:**
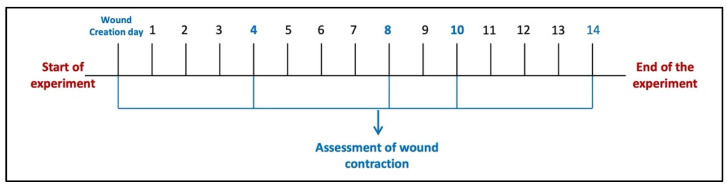
Diagram of the experimental protocol timeline.

**Table 1 pharmaceuticals-17-00733-t001:** Percentage of different fatty acids in Cucurbitaceae seed oils as detected in fatty acid methyl esters (FAMEs) analyzed by gas chromatography.

Name	Area Sum Percentage %			
	CSO	HSO	PSO	ZSO
Undecanoic acid	-	-	0.12	-
Lauric acid	1.59	0.18	0.29	0.20
Tridecanoic acid	1.88	0.23	0.29	0.23
Myristic acid	0.90	0.15	0.27	0.18
Palmitic acid	9.87	10.33	12.94	12.48
Palmitoleic acid	0.08	0.12	0.17	0.08
Stearic acid	5.11	5.28	7.96	7.72
Oleic acid	14.17	16.25	36.85	29.37
Linoleic acid	65.60	65.90	39.26	48.55
Linolenic acid	0.28	0.23	0.18	0.16
Arachidic acid	0.20	0.24	0.59	0.41
Behenic acid	0.10	0.05	0.18	0.12
Nervonic acid	0.07	0.91	0.24	0.32
Saturated fatty acids	19.65	16.46	22.64	21.34
Monounsaturated fatty acids	14.32	17.28	37.26	29.77
Polyunsaturated fatty acids	65.88	66.13	39.44	48.71
Total %	99.85	99.87	99.34	99.82

Reported percentages represent fatty acid methyl esters identified in the saponifiable fraction from CSO: cantaloupe seed oil, HSO: honeydew seed oil, PSO: pumpkin seed oil, and ZSO: zucchini seed oil.

**Table 2 pharmaceuticals-17-00733-t002:** The relative percentage of compounds in the unsaponifiable matter of Cucurbitaceae seed oils as analyzed by gas chromatography.

Name		Area Sum Percentage %		
	CSO	HSO	PSO	ZSO
Squalene	15.07	16.65	54.12	12.27
Longipinene epoxide	2.01	-	-	1.39
γ-Tocopherol	-	-	0.61	-
α-Tocopherol	-	-	0.34	-
Geranylgeranyl alcohol	9.14	13.23	-	0.92
21,25-Dihydroxy-vitamin D3	1.99	-	-	47.38
Stigmasterol	48.51	34.40	1.51	6.43
β-Sitosterol	7.88	14.89	14.88	9.59
Desmosterol	-	-	10.70	-
α-Amyrin	-	-	4.66	-
Simiarenol	-	-	7.65	-
Lupeol	-	-	0.30	-
Sterols	56.39	49.29	27.09	16.02
Terpenoidal compounds	26.22	29.88	66.73	14.58
Vitamins	1.99	-	0.95	47.38
%Total identified	84.59	79.17	94.77	77.99

Reported percentages represent compounds identified from the unsaponifiable fraction from CSO: cantaloupe seed oil, HSO: honeydew seed oil, PSO: pumpkin seed oil, and ZSO: zucchini seed oil.

**Table 3 pharmaceuticals-17-00733-t003:** Score of epidermal layer formation.

Group	Re-Epithelization	Epidermal Thickness
Normal control	2	2
Wound injury	1	0
Wound injury + PSO/ST	2	1
Wound injury + HSO	2	2
Wound injury + CSO	2	1
Wound injury + ZSO	2	2

Re-epithelization scoring (2, 1, and 0 indicate complete, partial, and none, respectively). Epidermal thickness scoring (2, 1, and 0 indicate normal, hypertrophy, and hypoplasia, respectively). CSO; cantaloupe seed oil, HSO; honeydew melon seed oil, PSO; pumpkin seed oil, ST; standard, ZSO; zucchini seed oil.

**Table 4 pharmaceuticals-17-00733-t004:** Primer sequences used for the investigated genes.

RAGE	forward: 5′-CAGGGTCACAGAAACCGG-3′ reverse: 5′-ATTCAGCTCTGCACGTTCCT-3′
*β*-actin	forward: 5′-AGGCCAACCGTGAAAAGATG 3′ reverse: 5′-ACCAGAGGCATACAGGGACAA3′

The primer sequences used for advanced glycation end products (RAGE) and β-actin (control for normalization).

## Data Availability

Data is contained within the article and Appendix A.

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
