# Peer review of "Wound Healing Efficacy of Cucurbitaceae Seed Oils in Rats: Comprehensive Phytochemical, Pharmacological, and Histological Studies Tackling AGE/RAGE and Nrf2/Ho-1 Cue"

_pharmaceuticals, 2024, doi:10.3390/ph17060733_

Round 1

Reviewer 1 Report

Comments and Suggestions for Authors

The study entitled “Wound healing efficacy of Cucurbitaceae seed oils in rats: comprehensive phytochemical, pharmacological, and histological studies, tackling AGE/RAGE and Nrf2/Ho-1 cue “is a very interesting study, presenting a substantial piece of scientific work, providing very interesting insights about comparative wound healing efficacy of certain Cucurbitaceae seed oils. Although the study seems to have very good aspects the manuscript shall be improved in order to better present this scientific work.

The abstract is well written describing the purpose as the possible use of an industrial waste for its therapeutic efficacy, but it is essential to describe most results into the abstract. Since we are dealing with a research article, we would need to be able to access its information at a glance. In order to improve clarity, the use of abbreviations in the abstract could be avoided.

The introduction must undergo extensive rewrite since it needs to focus to the Cucurbitaceae oils and wound healing. General description of integumentary system or other therapies or trends may also be removed. Furthermore, when describing the phases of wound healing it would be better to use the most accepted terminology (inflammatory phase, proliferative phase (which include reepithelization), and maturation - remodeling) although the description given in the text is not wrong.

At last, the purpose of the study must be well described and stated.  Is it the repurposing of industrial waste as ingredient of wound healing products as described in the abstract? Is it the fact the we need novel wound healing agents (because…..)? Is it something else? It is crucial to describe the motivation of the study in order to fulfill also ethical considerations about the use of laboratory animals.

Animals: The animals age shall be described. As seen, there is a big difference in weights (form the lowest to the highest weight aprox 20%). That means also that there was a substantial variation in the age. If it is the case, please provide information about randomization of the animals in groups in order to overpass this problem. Moreover, animals’ source and feeding status must be provided. Please check again your weight measurements because measuring round numbers for weight is extremely unusual. Why did you choose 6 animals as group size?

Anesthesia: The selection of phenobarbital as anesthetizing agent shall be justified (including reference) as it represents a less common method, having also a risk for animals’ welfare.

Wound infliction: The method of performing wound infliction must be better described. How were the researchers able to maintain the wounds size consistent? In the pictures wounds seem to be round or oval, did you use a biopsy punch?

Experimental protocol: The description of oil placement on the wounds shall be improved. It is stated that “Adhesive tape was applied softly to the skin to restrict any oil leakage”. The reader shall be informed about the nature and the type of adhesive tape as long as how it was used (directly to the wound? If it is the case, how do we know the tapes’ glue does not interfere with the wound healing process?). Was the same tape used to the controls also?  

The part of day zero shall be improved (it is better to state at start ecc)

The investigators probably were measuring the wound area by photo capture and not wound contraction. Although it is well established that wound healing in mice and rats involves wound contraction it is not the sole phenomenon during wound healing. Thus, the wound area reduction shall not be stated as wound contraction. Please report exactly what did you measure.

Please provide information why did you choose to end the experiment at the 14th day.

Tissue collection: Please provide information about animal’s sacrification mmeans. Was it performed according to AMVA or other guidelines? Please replace the word “scarification”

2.4.5 The devices used shall be stated

2.4.6 Although the phrase “the method described previously” is valid a brief description of the method may be included.

Table 1: Ideal tables shall be understandable also as stand-alone (without the text) thus legend must be rewritten

2.4.7 The ELISA meter shall be reported, methods of tissue treatment and extraction must be described

2.4.9 How were the results scored? Did you calculate total pixels stained? The method must be described.

2.5 Were the results tested for normality of distribution? How did you establish that the parametric test was the adequate for this analysis? Would you consider linear regression analysis to compare the different healing processes?

Table 2 Ideal tables shall be understandable also as stand-alone (without the text) thus table legend must be rewritten. Moreover it would be better to group fatty acids into ω-3 , ω-6 or ω-9  subgroups possibly providing total ω3 , ω6, ω9.

Table 3 Ideal tables shall be understandable also as stand-alone (without the text) thus table legend must be rewritten. Moreover, compounds would be better to be grouped in subcategories such as terpenes, sterols, vitamins ecc.

3.2 The term “wound contraction” is again confusing.  Figure 2 needs to be betterformated since photos are not of the same scale and seem to be deformed, please edit Figure 2

Histopathological evaluation: The subparagraph enumeration shall be changed (still 3.2). The Figure 3 needs to be edited in order that all sections have the sane orientation. This will facilitate the comparative assessment from readers. The magnification shall be stated. A scale bar may be added to improve readability. Consider implementing a hierarchical histological scoring table for histopathology assessment, although this cannot be suggested as mandatory.

Figure 4. Please check the statistics as for 3 samples measurement ANOVA is not the recommended method. Rather non parametric tests may me suitable. Please elucidate what do the Greek letters on the bars represent. The table also contains some results for β-actin that should be further explained.

Figure 5 The Graph could be more understandable, the Greek letters should be explained (better removed). Check the statistics, in order to use Parametric methods, you need to perform a normality test first.

Figure 6 The Graph could be more understandable, the Greek letters should be explained (better removed). Check the statistics, in order to use Parametric methods, you need to perform a normality test first.

Figure 7. Magnification must be reported. Scale bars may be added. Does the specimens represent the day after wound induction or the 14th day post wounding? Please correct the legend. Again, you need to see the statistics again. The number of specimens shall be reported. The Greek letters on the top of the columns what are they needed for?

Discussion: The first sentence may be chosen to be transferred to the introduction as it is well written. The first paragraph must explain what the researches did. It is obvious the agents used in the experimentation were used as topical agents and not nutraceuticals. Better rewrite the first paragraph.

The authors attribute the oils effectiveness to polyunsaturated fatty acid which is one part of their results. One other part of their results is a correlation of effectiveness with the content of Geranylgeranyl alcohol. This very important finding, in my opinion, may also be discussed knowing already from literature that some diterpenes have shown considerable effectiveness in wound healing.

Please avoid to state “wound contraction” because it is rather confusing.

The paragraph about Cxs in the discussion part needs better explanation and more referring to the results stating what the conclusion is at the end of the paragraph. Please rewrite the paragraph carefully.

Not sure if there is one seed oil that is better then the others since the research was performed in a very small group of animals and the results seem to be uncertain. The most important finding in my opinion is that all Cucurbitaceae seed oils seem to possess wound heling properties with small differences between them. Maybe the  oils with higher omega 3 content and higher diterpene content perform better but these are preliminary results not definite results.  

Please provide one or two paragraphs into your manuscript explaining the limitations of your study describing all the problems and limiting factors. The small sample size should be listed amongst them as one of the most important.

Comments on the Quality of English Language

Eglish language use is decent. 

Author Response

The authors appreciate the reviewer’s efforts for revising the manuscript entitled: Wound healing efficacy of Cucurbitaceae seed oils in rats: comprehensive phytochemical, pharmacological, and histological studies, tackling AGE/RAGE and Nrf2/Ho-1 cue”. Please find our responses to the reviewer’s comments in the uploaded file.

Reviewer 2 Report

Comments and Suggestions for Authors

t is incomplete, it is not mentioned if they used standards for the components of the oils, or how they determined the components present in the oils.

Author Response

The authors appreciate the reviewer’s efforts for revising the manuscript entitled: Wound healing efficacy of Cucurbitaceae seed oils in rats: comprehensive phytochemical, pharmacological, and histological studies, tackling AGE/RAGE and Nrf2/Ho-1 cue”. Please find below our responses to the reviewer’s comments.

Reviewer 3 Report

Comments and Suggestions for Authors

Dear authors,

Congratulations for the work, it is well structured and argued, however, I would like to make a few observations that will help increase the value of this work

1. All Latin scientific names must be written in italics! Check everywhere in the text.

2. In the Introduction Section, please give some examples of Plant-derived natural compounds that have the potential to enhace the wound healing process (as you mentioned)

3. In the Introduction Section you specified that these natural compunds are deemed safe in comparasion to synthetic substances. I think that the risks of natural products should also be highlighted, especially when they are used indiscriminately, because natural does not always mean safe.

4. Should the degree of novelty of the study be highlighted! What's new? These products have not been studied until now, from this point of view of usefulness in the treatment of wound healing? Arguments!

5. Please use a reference for the point 2.3.1. Sample derivatizations (at the end)...acording to which procedure this was done

6. For point 2.3.2, Gas chromatography analysis , you can complete with another reference such as the following

Robu S; Romila A; Buzia OD; Spac AF; Diaconu C; Tutunaru D; Lisa E; Nechita A. Contribution to the Optimization of a Gas Chromatographic Method by QbD Approach used for Analysis of Essential Oils from Salvia officinalis. REVISTA DE CHIMIE. 2019; 70(6): 2015-2020. ISSN: 0034-7752 WOS: 000475860100026

7. For in vivo protocol the place where the animals were purchased and acclimatized should be specified. 

8. For point 2.4.4. Tissue collection - animals sacrification

Good luck!

--

Author Response

(The authors gave the same response as above.)

Round 2

Reviewer 1 Report

Comments and Suggestions for Authors

The manuscript was improved substantially by the authors highlighting the excellent work performed, though some minor issues may be further addressed.

  1. The last part of the introduction must answer "why you did the experiment?" This is still missing.

  2. About the group size, there are still some uncertainties. In figure 2, you state that macroscopic assessment was performed only in 6 animals per group. How did you choose which 6 out of 9 you measured? It would be better to have all animals' measurements if possible. If not, please explain.

  3. In figure 4, although ANOVA performance is possible for 3 samples, in order to perform it, except for normality, you should check also for homogeneity of variances (Levene's test). Then you must provide in the paper the p-values obtained from Shapiro-Wilk and Levene. At last, you shall explain to the reader that the power of this test (ANOVA) is limited due to the low number of samples.

  4. CSO and HSO seem to have a tiny difference in omega-3 content. Authors must explain in the discussion their view about the difference in healing efficiency between CSO and HSO since they are attributing the healing potential of HSO to omega-3 PUFA.

Comments on the Quality of English Language

The quality of english language is decent although some small corrections can be made.  

Author Response

The authors appreciate the reviewer’s efforts for revising the manuscript entitled: Wound healing efficacy of Cucurbitaceae seed oils in rats: comprehensive phytochemical, pharmacological, and histological studies, tackling AGE/RAGE and Nrf2/Ho-1 cue”

Reviewer 3 Report

Comments and Suggestions for Authors

I am ok with the last form of the manuscript.

Author Response

(The authors gave the same response as above.)
